# Modulating Electron Density of Boron–Oxygen Groups in Borate via Metal Electronegativity for Propane Oxidative Dehydrogenation

**DOI:** 10.3390/ma17122868

**Published:** 2024-06-12

**Authors:** Panpan Li, Yongbin Yao, Shanshan Chai, Zhijian Li, Fan Xue, Xi Wang

**Affiliations:** 1Key Laboratory of Luminescence and Optical Information, Ministry of Education, School of Physical Science and Engineering, Beijing Jiaotong University, Beijing 100044, China; 19118030@bjtu.edu.cn (P.L.); yyb0321shzu@163.com (Y.Y.); 20118011@bjtu.edu.cn (S.C.); 23111524@bjtu.edu.cn (Z.L.); 22121615@bjtu.edu.cn (F.X.); 2Tangshan Research Institute of Beijing Jiaotong University, Tangshan 063000, China

**Keywords:** electronegativity, borate, propane oxidative dehydrogenation, [BO_3_]^3−^ group

## Abstract

The robust electronegativity of the [BO_3_]^3−^ structure enables the extraction of electrons from adjacent metals, offering a strategy for modulating oxygen activation in propane oxidative dehydrogenation. Metals (Ni 1.91, Al 1.5, and Ca 1.0) with varying electronegativities were employed to engineer borate catalysts. Metals in borate lacked intrinsic catalytic activity for propane conversion; instead, they modulated [BO_3_]^3−^ group reactivity through adjustments in electron density. Moderate metal electronegativity favored propane oxidative dehydrogenation to propylene, whereas excessively low electronegativity led to propane overoxidation to carbon dioxide. Aluminum, with moderate electronegativity, demonstrated optimal performance. Catalyst AlBOx-1000 achieved a propane conversion of 47.5%, with the highest propylene yield of 30.89% at 550 °C, and a total olefin yield of 51.51% with a 58.92% propane conversion at 575 °C. Furthermore, the stable borate structure prevents boron element loss in harsh conditions and holds promise for industrial-scale catalysis.

## 1. Introduction

The efficient activation of oxygen molecules stands as a pivotal step in the oxidative dehydrogenation of propane (ODHP) reaction [1,2,3]. Research indicates that oxygen-containing functional groups in boron-based catalysts, including BO, B(OH)_x_O_3−x_ (x = 0–3), and B-O-B structures, serve as the active sites for oxidative dehydrogenation of propane [4,5,6]. These active sites interact with oxygen molecules, transforming them into active oxygen species, thereby facilitating the cleavage of C-H bonds in propane molecules and subsequent production of propylene [2]. Therefore, meticulous control over these active sites to bolster their oxygen activation capability emerges as a promising strategy for the design and development of efficient catalysts for propane oxidative dehydrogenation [2].

Borate materials have garnered extensive applications in fields such as nonlinear optics and laser technology due to their distinctive structural properties [7,8,9]. Various coordination modes between boron and oxygen atoms exist in borate structures, leading to the formation of planar triangular [BO_3_]^3−^ structures and tetrahedral [BO_4_]^5−^ structures. These fundamental units can further amalgamate into a spectrum of chain, layered, or three-dimensional network structures by sharing oxygen atoms [10,11,12]. Consequently, the coexistence of multiple boron–oxygen units in borates facilitates the presence of active sites for catalyzing propane oxidative dehydrogenation [13]. Qian et al. [14] introduced borates as catalysts for the oxidative dehydrogenation of propane, and demonstrated remarkable selectivity towards olefins. The ^11^B NMR spectrum revealed that these three-coordinated boron–oxygen ring groups were active sites. The strong electronegativity of the [BO_3_]^3−^ structure could elevate the electron density of the [BO_3_]^3−^ group via extracted electrons from the adjacent metal, thereby enhancing its oxygen activation capability [15,16]. Hence, manipulation of the electron density on the [BO_3_]^3−^ group can be utilized to regulate the performance of the propane oxidative dehydrogenation reaction.

In this study, we engineered and synthesized the metal borate catalyst MBOx (M = Ni, Al, Ca), and regulated the catalytic performance through the manipulation of the electron density of boron–oxygen functional groups, and the manipulation realized via the differing electronegativities between the metals and [BO_3_]^3−^ groups. Metals (Ni 1.91, Al 1.5, and Ca 1.0) with varying electronegativities were employed to engineer borate catalysts. Our research results indicated that the metals in borate lacked intrinsic catalytic activity for propane conversion; instead, they modulated [BO_3_]^3−^ group reactivity through adjustments in electron density. Moderate metal electronegativity favored propane oxidative dehydrogenation to propylene, whereas excessively low electronegativity led to propane overoxidation to carbon dioxide. Aluminum (Al) demonstrated a moderate electronegativity among the three metals, and the catalyst AlBOx-1000 achieved optimal performance in propane oxidative dehydrogenation. Catalyst AlBOx-1000 achieved a propane conversion of 47.5%, with the highest propylene yield of 30.89% at 550 °C, and a total olefin yield of 51.51% with a 58.92% propane conversion at 575 °C. Furthermore, the stable borate structure prevents boron element loss in harsh conditions and holds promise for industrial-scale catalysis.

## 2. Materials and Methods

### 2.1. Catalyst Preparation

The Pechini method is a widely employed synthesis technique, wherein metal ions engage in the formation of coordination complexes with α-hydroxy acids [17,18]. Subsequently, these complexes undergo a reaction with polyhydroxy alcohols under heating conditions, leading to the formation of cross-linked networks. This process is followed by additional heating to facilitate water removal, culminating in the production of solid polymers. In the preparation of borate precursors, the Pechini method was employed by dissolving specific quantities of metal salt, boric acid, and citric acid in water, followed by the addition of ethylene glycol as a chelating agent (All reagents are purchased from Aladdin Co., Ltd., Fukuoka, Japan). The resulting mixture underwent heating in an oil bath to facilitate thorough cross-linking between metal–citrate complexes and ethylene glycol, followed by further heating to eliminate water, yielding a solid precursor. The prescribed molar ratios were 1:1 for metal salt to boric acid, 1:2 for metal salt to citric acid, and 1:1 for ethylene glycol to citric acid.

The preparation of borate involved subjecting the borate precursor to roasting in a muffle furnace, considering its high organic content, the decomposition was addressed in three stages: 

(a) Initially, the precursor was roasted at 300 °C for 2 h with a heating rate of 1 °C/min. Following this, the sample was allowed to cool to room temperature before further utilization.

(b) Subsequently, the powder sample obtained from stage one was roasted at 650 °C for 4 h with a heating rate of 2 °C/min. After roasting, the sample was cooled to room temperature, ground, and prepared for subsequent use. This resulting sample was designated as MBOx-650 (M = Al, Ni). 

(c) Lastly, the powder sample obtained from stage two underwent roasting at 1000 °C for 4 h with a heating rate of 10 °C/min.

### 2.2. Characterization

X-ray diffraction (XRD) experiments were carried out using a Rigaku D/Max-2500 (Rigaku Co., Ltd., Tokyo, Japan) instrument, featuring a Cu Kα (0.15406 nm) radiation source, variable slits, and a Shimadzu detector. The XRD measurements operated at 100 mA and 40 kV, with parameters set at 2θ = 10–80° and a scan rate of 5.0°/min. Fourier transform infrared spectra (FT-IR) were obtained using a BRUKER TENSOR-27 (Brucker Co., Ltd., Karlsruhe, Germany) infrared spectrometer with a resolution of 4 cm^−1^. X-ray photoelectron spectroscopy (XPS) analysis was carried out using an ESCALAB250XI instrument ((VG Co., Ltd., Park Ridge, IL, USA), equipped with a monochromatic Al Kα X-ray source (energy: 1486.68 eV). A SU-8010 (Hitachi Co., Ltd., Tokyo, Japan) field-emission scanning electron microscope (FE-SEM) was employed for morphological analysis of samples and elemental distribution.

### 2.3. Catalytic Performance Evaluation

The evaluation of the propane oxidative dehydrogenation performance of borate catalysts was performed as follows: Initially, the borate catalyst underwent compression and grinding into pellets, following which 400 mg of the catalyst was introduced into a quartz tube for performance evaluation. The feed gas composition comprised propane and oxygen in a 1:2 ratio, with nitrogen employed as the balance gas. The flow rate during testing was maintained at 2400 L kgcat^−1^ h^−1^. For direct propane dehydrogenation testing, the feed gas composition maintained a 1:3 ratio of propane to nitrogen, while all other conditions remained constant.

Gas product analysis was performed utilizing a gas chromatograph manufactured by Panna (Changzhou) Instruments Co., Ltd. (Changzhou, China), model A91. Organic compounds were detected employing a flame ionization detector (FID), coupled with a capillary column from Agilent (Santa Clara, CA, USA), HP-AL/S, featuring dimensions of 25 m × 0.32 mm × 8 μm. Furthermore, the analysis of inorganic gases was conducted employing a thermal conductivity detector (TCD).

## 3. Results and Discussion

### 3.1. Design and Synthesis of Borate Catalyst

The inherent high electronegativity of [BO_3_]^3−^ groups facilitated the abstraction of electrons from neighboring metal atoms, offering a strategy to modulate the electron density of [BO_3_]^3−^ groups via the diverse electronegativities of the metal atoms. Considering the varied atomic electronegativities of different metals [19,20], nickel (Ni, 1.91), aluminum (Al, 1.5), and calcium (Ca, 1.0) were selected as metal constituents to synthesize the corresponding borate catalysts MBOx (M = Ni, Al, Ca), enabling precise manipulation of the electron density on [BO_3_]^3−^ groups (Figure 1a). The synthesis of catalysts borate MBOx (M = Ni, Al, Ca), was illustrated in Figure 1b, involved the initial dissolution of metal salts in an aqueous solution containing boric acid and citric acid, resulting in the formation of coordination complexes. Subsequent reaction with ethylene glycol under elevated temperatures yielded a cross-linked network structure, followed by additional heating to remove water and obtain a solid polymer. Ultimately, the corresponding borate catalysts MBOx (M = Ni, Al, Ca) were achieved through a calcination process.

We performed comprehensive morphology characterization of the synthesized borate catalysts MBOx (M = Ni, Al, Ca). Scanning electron microscopy (SEM) analysis revealed that the catalysts AlBOx-1000 and NiBOx-1000 showed rod-like morphologies, whereas the catalyst CaBOx-1000 exhibited a typical lamellar structure. These morphological characteristics are depicted in Figure 1c–e. Furthermore, utilizing the energy-dispersive X-ray spectroscopy (EDS) mapping technique, we verified the homogeneous distribution of Ni (or Al or Ca), B, and O elements within the borate catalysts (Appendix A). These findings affirmed the successful synthesis of borate catalysts MBOx (M = Ni, Al, Ca).

### 3.2. Characterization of Borate Structure

To understand the structural properties of borate catalysts MBOx (M = Ni, Al, Ca), a series of characterization was conducted. Initially, we performed X-ray diffraction analysis. Catalyst NiBOx-1000 exhibited diffraction peaks observed at 2θ = 22.620°, 25.900°, 33.178°, 33.839°, 36.476°, 40.531°, 41.891°, 53.256°, 55.572°, etc. (Figure 2a, top). Extensive analysis employing MDI Jade 6 software confirmed that these peaks correspond to Ni_3_(BO_3_)_2_ (JCPDS: 22-0745) [21]. Catalyst NiBOx-650 demonstrated analogical X-ray diffraction profiles with inferior crystallinity in comparison to catalyst NiBOx-1000 (Appendix A). Notably, catalysts NiBOx-1000 and NiBOx-650 exhibited diffraction peaks of NiO at 2θ = 37.248°, 43.275°, 62.878°, 75.414°, 79.407°, indicating the persistence of NiO peaks despite the elevated calcination temperature of NiBOx-1000, possibly attributable to incomplete conversion into oxide Ni_3_(BO_3_)_2_ during the calcination process. Catalyst AlBOx-1000 manifested distinct diffraction peaks at 2θ = 16.486°, 20.321°, 23.698°, 26.402°, 33.326°, 35.788°, 39.822°, etc. (Figure 2a, middle). Subsequent analysis authenticated these peaks as indicative of Al_20_B_4_O_36_ (JCPDS: 80-2300) [22]. Conversely, a broad peak within the 20–30° range was singularly displayed in the diffraction pattern of catalyst AlBOx-650, indicative of an amorphous state during the 650 °C calcination phase (Appendix A). Additionally, minor peaks attributed to boron aluminum functional groups were identified upon further investigation. Distinct diffraction peaks of catalyst CaBOx-1000 emerged at 2θ = 28.870°, 30.807°, 32.411°, 35.743°, 41.784°, 45.788°, 48.103°, etc. (Figure 2a, bottom); these peaks were conclusively identified to correspond to Ca_2_B_2_O_5_ (JCPDS: 28-0209) via meticulous analysis. Furthermore, supplementary minor peaks were discerned and subsequently determined to align with Ca_2_B_2_O_5_ (JCPDS: 18-0279) and CaB_2_O_4_ (JCPDS: 32-0155). Additionally, catalyst CaBOx-650 exhibited distinct diffraction peaks compared to catalyst CaBOx-1000, with a relatively diminished intensity of peaks (Appendix A). The present study demonstrated the successful synthesis of borate catalysts employing the citric acid method, and elucidated the significant influence of calcination temperature on the crystalline phase of such catalysts.

To gain further insights into the structure of the catalyst, we conducted characterization of catalysts employing Fourier transform infrared (FTIR) spectroscopy (Figure 2e), and generated structural diagrams of these catalysts by integrating X-ray diffraction data with Diamond (version number 3.2) software (Figure 2b–d). In catalyst NiBOx-1000, B atoms formed planar BO_3_ structures with three oxygen atoms, while Ni atoms formed octahedral structures with six O atoms within the crystal lattice of Ni_3_(BO_3_)_2_ (Figure 2b and Appendix A). These observations were validated by FTIR spectra, which exhibited prominent Ni-O vibrations [23] between 600–723 cm^−1^ and BO_3_ bond vibrations between 1200–1500 cm^−1^ [14,24], and a distinct peak at 723 cm^−1^ corresponded to -OH vibrations (Figure 2e, red). The structural diagram of AlBOx-1000 revealed two distinct forms of Al atoms: one formed an octahedral structure with six oxygen atoms, while the other exhibited five-coordinated structures with five oxygen atoms (Figure 2c and Appendix A). Vibration bands spanning 500–1000 cm^−1^ corresponded to A-O and Al-O-Al bond vibrations (Figure 2e, blue) [25,26], and vibrations within the 1200–1500 cm^−1^ range aligned with BO_3_ bond vibrations as depicted in the structural diagram [14,24]. Furthermore, vibrations attributed to Al-OH (3450 cm^−1^) and B-OH (3230 cm^−1^) were detected within the 3000–3500 cm^−1^ range. Additionally, BO_3_ bond vibrations within the 1200–1500 cm^−1^ range corroborated the depiction of B atoms forming planar BO_3_ structures with three oxygen atoms in catalyst CaBOx-1000 (Figure 2d,e, green). Vibration bands within the 600–1020 cm^−1^ range were indicative of Ca-O bond vibrations [27,28,29], in accordance with the distorted octahedral structure formed by Ca coordinated with seven O atoms.

Planar BO_3_ structures presented in all borate catalysts MBOx (M = Ni, Al, Ca), and the pronounced electronegativity of [BO_3_]^3−^ groups enabled its acquisition of electrons from adjacent metal atoms. Consequently, the disparate electronegativities inherent to the metal constituents within the borate catalysts MBOx (M = Ca, Al, Ni) engendered diverse electron densities of the [BO_3_]^3−^ groups. Catalyst NiBOx-650 demonstrated analogical X-ray diffraction profiles corresponding to Ni_3_(BO_3_)_2_ (JCPDS: 22-0745) with inferior crystallinity in comparison to catalyst NiBOx-1000. Nevertheless, heightened crystallinity led to a decrease in the binding energies of B 1s and O 1s, potentially attributable to the improved crystalline structure facilitating the extraction of electrons by the [BO_3_]^3−^ groups from neighboring Ni metals (Figure 2f). Although the loss of electrons typically led to an increase in binding energy in metal atoms, an anomalous downward shift in the binding energy of Ni 2p was observed (Appendix A). This phenomenon could potentially be ascribed to the formation of covalent bonds between nickel and oxygen atoms. Such bonding promoted hybridization between the d orbitals of nickel and the p orbitals of oxygen, facilitating electron sharing and the generation of new molecular orbitals. Consequently, the binding energy of nickel exhibited a shift towards lower values [30]. The B 1s and O 1s binding energies of catalyst CaBOx-1000 showed a similar tendency to catalyst NiBOx-1000, thereby implying an augmented electron density within the [BO_3_]^3−^ groups (Appendix A). However, due to the amorphous nature of catalyst AlBOx-650, it was impossible to directly infer an increase in electron density on the functional group by simply comparing the binding energies of elements (Appendix A).

### 3.3. Catalytic Performance 

Considering the ubiquitous role of tricoordinate boron (BO_3_) as pivotal catalytic sites for propane oxidative dehydrogenation (OPDH), it was reasonable to anticipate that all three catalysts could facilitate propane conversion. This variance of [BO_3_]^3−^ groups substantially modulated the activation proficiency of oxygen molecules, thereby exerting a consequential impact on the efficacy of propane oxidation dehydrogenation. To investigate the influence of various metals on the propane oxidation dehydrogenation reaction, we evaluated catalytic performance and subsequently delineated the outcomes individually.

Catalyst NiBOx-1000 achieved a propane conversion of 4.47% at 450 °C, and the propane conversion escalated swiftly with temperature, nearly doubling with every 25 °C increment below 550 °C (Figure 3a). The propane conversion attained 42.3% at 550 °C, yielding a total olefin output of 38.84%, and the propane conversion climbed to 57.06% at 575 °C. The selectivity towards propylene diminished with the increasement of propane conversion concurrently, particularly beyond 550 °C, where propylene selectivity experienced a rapid decline (Figure 3b). Despite the escalation of side reactions with increasing temperature, ethylene, a by-product with relatively high added value, suggested that the catalyst NiBOx-1000 mitigated the profound dehydrogenation oxidation of propane (Figure 3c). Catalyst NiBOx-650 demonstrated relatively inferior catalytic performance in propane oxidation dehydrogenation over the entire temperature range, and the propane conversion of catalyst NiBOx-650 was merely 15.5% at 575 °C (Appendix A). This discovery implied that the calcination temperature exerted a substantial influence on the catalytic performance of the catalyst, possibly due to the higher calcination temperatures leading to improved crystallinity of the catalyst. This improved crystallinity facilitated effective electron transfer, consequently augmenting the activation of oxygen molecules.

Compared to nickel (1.91), aluminum (1.5) has a lower electronegativity, making it easier for the highly electronegative [BO_3_]^3−^ groups to gain electrons from aluminum. Therefore, the catalyst AlBOx-1000 was anticipated to demonstrate superior performance in propane oxidation dehydrogenation processes. Experimental results have indeed shown that catalyst AlBOx-1000 demonstrated superior performance in propane oxidation dehydrogenation throughout the entirety of the reaction temperature range compared to catalyst NiBOx-1000 (Figure 3a). The propane conversion on catalyst AlBOx-1000 steadily rose with increasing reaction temperature, approximately doubling per 25 °C increment below 575 °C, while propylene selectivity marginally decreased below 500 °C, and sharply declined beyond 525 °C. However, the overall olefin selectivity experienced a modest decrease, not exceeding 10% across the entire tested temperature range (Figure 3c). At 550 °C, the catalyst AlBOx-1000 achieved a propane conversion of 47.5%, and the total olefin selectivity remained 90.87% despite the propylene selectivity notably decreased at 550 °C (65.1%). The propane conversion reached 58.92%, with total olefin selectivity at 87.42%, resulting in a total olefin yield of 51.51% at 575 °C. Furthermore, the overall olefin yield of the catalyst AlBOx-1000 exceeded the catalyst AlB_2_ developed by our research team (30.2% at 500 °C) [31]. The catalyst AlBOx-650, with its amorphous structure, displayed relatively subdued catalytic activity, with propane conversion gradually escalating with temperature, reaching the highest conversion of 26.2% at 575 °C (Appendix A).

Despite calcium (1.0) being characterized by the lowest electronegativity, the utilization of calcium borate as a catalyst for propane oxidative dehydrogenation manifested suboptimal performance (Figure 3a). Upon reaching 475 °C, propane underwent complete conversion to carbon dioxide (Figure 3b,c). This phenomenon may be attributable to the increased electron density on the [BO_3_]^3−^ groups, which facilitated the activation of oxygen into more potent oxidative species. Additionally, we performed N_2_ adsorption–desorption isotherms for the three catalysts and measured their specific surface areas (Appendix A). There were no significant differences in the specific surface areas of the three catalysts. The differences in catalytic performance were primarily due to the electronegativity of the metals. Based on the structure and catalytic performance of the catalyst, we revealed that the electronegativity of metal atoms has a direct impact on the propane oxidation dehydrogenation performance of the catalyst. To substantiate the role of metal atoms in the propane oxidation dehydrogenation reaction, we carried out further performance evaluations of the catalyst.

### 3.4. Exploration of Active Sites

Nickel has garnered considerable attention as a catalyst in both propane dehydrogenation and propane oxidative dehydrogenation. The structural sensitivity of the propane dehydrogenation process has been shown to involve a solitary active atom for efficient catalysis, whereas C-C bond cleavage predominantly has occurred with multi-atom metal sites [32,33,34]. When nickel has been employed as a catalyst for propane oxidative dehydrogenation, its pronounced oxidative tendencies have rendered it prone to exacerbating propane over-oxidation, and consequently diminished olefin selectivity [35,36,37]. Therefore, meticulous modulation of nickel-based catalysts involved configuring them as either single-atom or alloys integrated with inert additives to preempt undesired side reactions.

We evaluated the catalytic performance of the catalyst NiBOx-1000 in propane dehydrogenation reactions, with a comparison of its performance in propane oxidative dehydrogenation depicted in Figure 4a and Appendix A. Catalyst NiBOx-1000 manifested inadequate propane dehydrogenation catalytic performance across the temperature range 450–575 °C, and achieved the optimal propane conversion of 2.71% at 575 °C. Such an observation substantiated that the Ni sites within the catalyst NiBOx-1000 demonstrated an intrinsic inability to catalyze the activation of propane molecules. Resembling the catalyst NiBOx-1000, catalyst CaBOx-1000 demonstrated inferior catalytic performance in the propane dehydrogenation reaction (Figure 4b and Appendix A). This finding implied that the metal atoms within borate catalysts lacked the intrinsic capability to catalyze propane conversion. Instead, their role involved the formation of stable crystal configurations with [BO_3_]^3−^ groups and the modulation of [BO_3_]^3−^ group activity through adjustments in electron density.

To gain a more profound understanding of the factors contributing to the heightened olefin selectivity in borate catalysts, nickel oxide (NiO) and B_2_O_3_ were employed and the performance of propane oxidative dehydrogenation was evaluated. The propane conversion of NiO was merely 1.66% at 450 °C, contrasting with the 4.47% propane conversion observed with catalyst NiBOx-1000, indicating superior activity of the (BO_3_)-Ni_3_- sites at lower temperatures (Figure 4c,d). The propane conversion soared to 66.95% at 475 °C with NiO, but showed negligible olefin selectivity (0.2%), with the primary product being CO_2_. The propane underwent complete conversion to CO_2_ above 475 °C, whereas the catalyst NiBOx-1000 maintained superior olefin selectivity throughout the temperature, underscoring the ability of (BO_3_)-Ni_3_- sites to suppress the extensive oxidative dehydrogenation of propane. The bonding between Ni and O was predominantly governed by ionic interactions in NiO; nonetheless, Ni and O atoms predominantly engage in covalent bonding in the catalyst NiBOx-1000. The C_3_H_7_• generated during the reaction process was electron-deficient and tended to adsorb onto electron-rich boron oxygen groups. Consequently, over-oxidative dehydrogenation of propane molecules was suppressed on the catalyst, leading to a higher olefin selectivity. B_2_O_3_ exhibited superior propene conversion and selectivity compared to the catalyst NiBOx-1000 within the temperature range of 450–500 °C (Appendix A). Nevertheless, its propene selectivity exhibited a more pronounced decrease compared to that of the catalyst NiBOx-1000 at temperatures exceeding 500 °C. B_2_O_3_ displayed a propylene selectivity of merely 34.03%, whereas catalyst NiBOx-1000 surpassed 68%, with an overall olefin selectivity surpassing that of B_2_O_3_. Additionally, there was a significant loss of the element B from B_2_O_3_ during the reaction process, which led to a noticeable reduction in catalyst volume and its transformation into a glassy state. No obvious change was observed in catalyst NiBOx-1000, and subsequent X-ray diffraction analysis of the spent catalysts revealed an absence of discernible alterations in diffraction patterns, thereby attesting to the stability of catalyst MBOx (M = Ni, Al, Ca) (Appendix A).

### 3.5. Reaction Mechanism

The reaction mechanism on borate catalysts primarily adhered to an oxidation-reduction mechanism, commonly referred to as the Mars–Van Krevelen mechanism [38,39]. Accordingly, we proposed a plausible reaction mechanism on the catalyst (Figure 5): Initially, propene molecules adsorbed onto the O atoms of (BO_3_)-Ni_3_-sites, followed by the abstraction of a hydrogen atom from the secondary carbon by a neighboring oxygen atom, resulting in the formation of a hydroxyl group and a C_3_H_7_•. Subsequently, a hydrogen atom from the first carbon was abstracted by another adjacent oxygen, facilitating the desorption of the propene molecule. The hydroxyl groups of two molecules condensed to form one molecule of water, which desorbed, leaving behind an oxygen vacancy. Finally, the oxygen vacancy was replenished by oxygen from the gas phase, thereby restoring the active site and completing the cycle. The C_3_H_7_• generated during the reaction process was electron deficient and tended to be adsorbed onto electron-rich boron oxygen groups. Consequently, over-oxidative dehydrogenation of propane molecules was suppressed on the catalyst, leading to a higher olefin selectivity.

## 4. Conclusions

In this study, we engineered metal borate catalysts MBOx (M = Ni, Al, Ca), and manipulated their catalytic performance by regulating the electron density of boron–oxygen functional groups. This manipulation was realized via leveraging the varying electronegativities between the metals and [BO_3_]^3−^ groups. The synthesis of the metal borate catalyst was accomplished using the citric acid method under relatively benign reaction conditions. Metals (Ni 1.91, Al 1.5, and Ca 1.0) with varying electronegativities were employed, and oxygen activation capability increased with decreasing electronegativity of metals. Metals in borate lacked intrinsic catalytic activity for propane conversion; instead, they modulated [BO_3_]^3−^ group reactivity through adjustments in electron density. Moderate metal electronegativity favored propane oxidative dehydrogenation to propylene, whereas excessively low electronegativity led to propane overoxidation to carbon dioxide. Aluminum, with a moderate electronegativity, demonstrated optimal performance. Catalyst AlBOx-1000 achieved a propane conversion of 47.5%, with the highest propylene yield of 30.89% at 550 °C, and a total olefin yield of 51.51% with a 58.92% propane conversion at 575 °C. Catalyst NiBOx-1000 exhibited slightly inferior catalytic performance compared to catalyst AlBOx-1000. Calcium (1.0), with the lowest electronegativity, facilitated the activation of oxygen into more potent oxidative species, leading to complete propane conversion to carbon dioxide. The stable borate structure effectively mitigated potential boron element loss in high-temperature and humid conditions. Consequently, borate catalysts exhibit potential for industrial-scale reactions.

## Figures and Tables

**Figure 1 materials-17-02868-f001:**
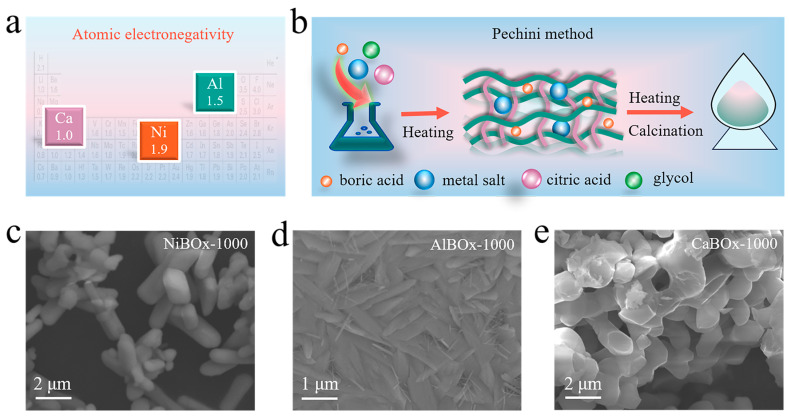
(**a**) Periodic table of electronegativities; (**b**) Synthesis schematic of borate catalysts MBOx (M = Ni, Al, Ca); scanning electron microscope (SEM) images of (**c**) catalyst NiBOx-1000, (**d**) catalyst AlBOx-1000, (**e**) catalyst CaBOx-1000.

**Figure 2 materials-17-02868-f002:**
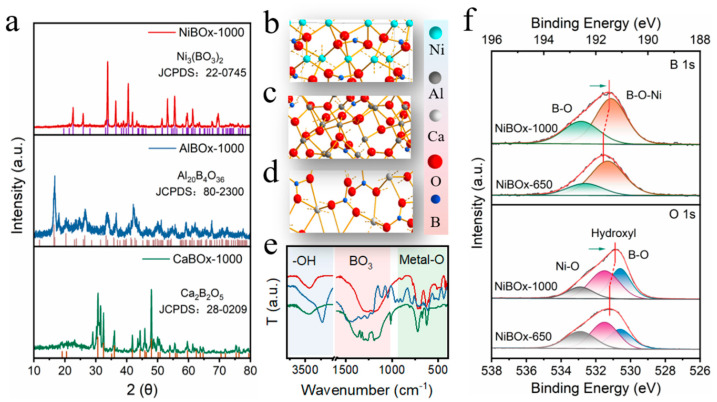
(**a**) X-ray diffraction (XRD) spectrum of borate catalysts MBOx (M = Ni, Al, Ca) and the corresponding structure generated by Diamond software (**b**–**d**); (**e**) Fourier transform infrared (FTIR) spectrum of borate catalysts MBOx (M = Ni, Al, Ca); (**f**) B 1s and Ni 2p X-ray photoelectron spectroscopy (XPS) of borate catalyst NiBOx-1000.

**Figure 3 materials-17-02868-f003:**
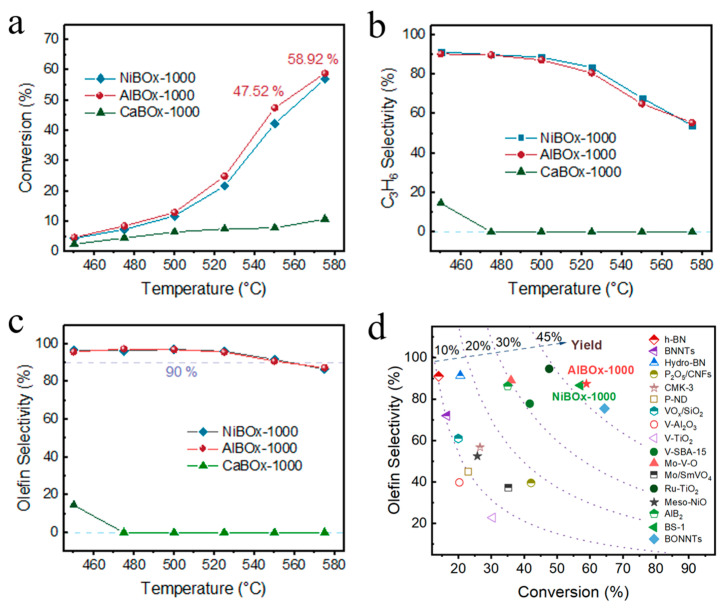
Propane oxidative dehydrogenation performance of borate catalysts MBOx (M = Ca, Al, Ni), (**a**) Propane conversion, (**b**) C_3_H_6_ selectivity, (**c**) Olefin selectivity, (**d**) Comparison of catalysts NiBOx-1000 and AlBOx-1000 with the established OPDH catalysts at their maximum reported yield.

**Figure 4 materials-17-02868-f004:**
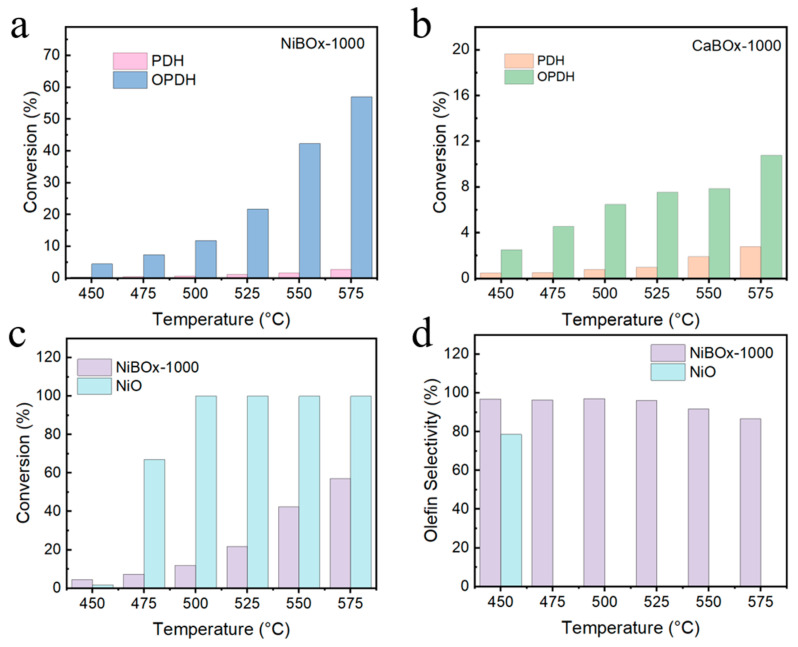
Catalytic performance comparison of propane oxidative dehydrogenation and direct dehydrogenation with the catalysts (**a**) NiBOx-1000 and (**b**) CaBOx-1000; Propane oxidative dehydrogenation performance of catalysts NiBOx-1000 and NiO: (**c**) Propane conversion, (**d**) Olefin selectivity.

**Figure 5 materials-17-02868-f005:**
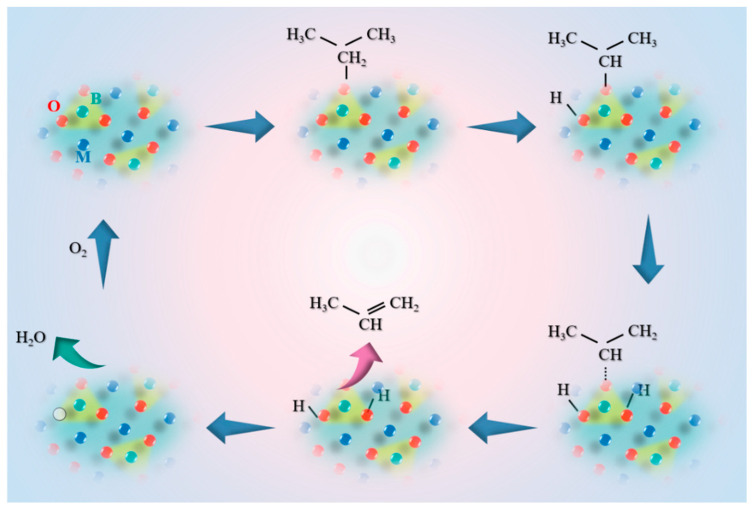
Possible reaction mechanism on borate catalyst, B green, O red, Ni blue.

## Data Availability

Data are contained within the article.

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
