# Peer review of "Modulating Electron Density of Boron–Oxygen Groups in Borate via Metal Electronegativity for Propane Oxidative Dehydrogenation"

_materials, 2024, doi:10.3390/ma17122868_

Round 1

Reviewer 1 Report

Comments and Suggestions for Authors

The authors describe the preparation of metal borates for the oxidative dehydrogenation of propane. The activity of the central borate element was tuned by incorporating metal atoms (Ni, Al, Ca) of different electronegativity, which enabled to adjust the activity of the catalyst in terms of propane conversion and propylene yield. This is an interesting piece of work; several points need to be considered before publication :

Figure 2f presents the B 1s and O 1s, while the figure caption mentions the B 1s and Ni 2p XPS region.

The authors discuss some shifts of XPS BE of the catalysts. What are the reference BE taken to account for a presumably significant shift? It seems that the mentioned shifts refer to shifts linked with the different temperature of calcination (650 vs 1000°C) and not the integration of the metal atoms to the borate structure itself.

For conversions, selectivities and yields, numbers should be given with maximum 3 digits; using more than one decimal make no sense here.

From the presented data, it seems that AlBOx-1000 and NiBOx-1000 have very similar activity, (much superior to that of CaBOx-1000), making difficult to select between Al and Ni. Are the tests performed in replicates to evaluate standard deviations on the experimental results, and objectively select between Al and Ni?

The authors compare the AlBOx-1000 catalyst with AlB2 and state that the former depict superior catalytic performances. AlB2 presents an overall olefin yield of 30.2% at 500°C, however at this temperature (500°C) and with the AlBOx-1000, the propane conversion is barely 10% with an olefin selectivity of ca. 90 %, which gives an olefin yield of less than 10 % - which is much smaller to that of AlB2 at the same temperature – assuming also that both tests were performed under the same reaction conditions (loading, gas flo rate, O2 content, etc.)…

The CaBOx-1000 catalyst showed exceedingly low PODH activity, if any, yielding to overoxidation of propane to CO2. Can structural/morphological effects, e.g. the size of the calcium atom (vs Ni and Al), complement the explanations based on electronics. In this regard, textural properties should also be measured for the 3 shaped (pellets) catalysts (N2-physisorption) as used for the catalytic tests

Figure 3d is not discussed

Figure 4c and 4d obviously refer to the OPDH reaction? to be mentioned in the figure caption

Figure 5 is difficult to comprehend. The authors should draw their tentative reaction mechanism using a central metalloborate element and draw the different reaction steps from propane, together with naming the different steps in link with the description in the main text – this using proper chemical structures

Overall, the english phrasing should be improved

Comments on the Quality of English Language

Some english phrasings need to be improved

Author Response

Manuscript ID: materials-3028063

Title: Modulating Electron Density of Boron-Oxygen Groups in Borate via Metal Electronegativity for Propane Oxidative Dehydrogenation

Authors: Panpan Li, Yongbin Yao, Shanshan Chai, Zhijian Li, Fan Xue, Xi Wang *

Dear Ms. Diana Alexandra Minea,

Thank you very much for your editorial comments on our manuscript. Firstly, we sincerely appreciate the professional and valuable evaluations and suggestions provided by the editor and reviewers for our work. The insightful and expert comments from the two reviewers have significantly contributed to the enhancement of our work.

We have addressed the comments point by point below. All changes in the revised manuscript were noted with red font. If further information is required, please do not hesitate to contact us.

We hope this manuscript will meet your publication criteria.

Yours sincerely,

Xi Wang

Reviewer 1:

The authors describe the preparation of metal borates for the oxidative dehydrogenation of propane. The activity of the central borate element was tuned by incorporating metal atoms (Ni, Al, Ca) of different electronegativity, which enabled to adjust the activity of the catalyst in terms of propane conversion and propylene yield. This is an interesting piece of work; several points need to be considered before publication:

  1. Figure 2f presents the B 1s and O 1s, while the figure caption mentions the B 1s and Ni 2p XPS region.

Response: Thank you very much for pointing out our mistakes. We have made the necessary corrections in the manuscript.

  1. The authors discuss some shifts of XPS BE of the catalysts. What are the reference BE taken to account for a presumably significant shift? It seems that the mentioned shifts refer to shifts linked with the different temperature of calcination (650 vs 1000°C) and not the integration of the metal atoms to the borate structure itself.

Response: Thank you very much for your suggestions. It is challenging to directly determine changes in electron density on the [BO3]3- group in catalyst NiBOx-1000 based on current characterization. In general, intensified interatomic interactions can augment the binding energy of XPS peaks in highly crystalline samples. However, on the highly crystalline catalyst NiBOx-1000, the binding energy of B1s shifts towards lower binding energy. The presence of the excess electron density in B vicinity is expected to perturb the binding strength of 1s electron and shift the bind energy to a lower value (Wen-Duo Lu et al. Chinese Journal of Catalysis 41 (2020) 1837–1845).” This indicates that there is an increase in electrons on the [BO3]3- group indirectly.

  1. For conversions, selectivities and yields, numbers should be given with maximum 3 digits; using more than one decimal make no sense here.

Response: Thank you very much for your suggestions. We have made the corresponding changes and replaced the images in the manuscript.

  1. From the presented data, it seems that AlBOx-1000 and NiBOx-1000 have very similar activity, (much superior to that of CaBOx-1000), making difficult to select between Al and Ni. Are the tests performed in replicates to evaluate standard deviations on the experimental results,and objectively select between Al and Ni?

Response: Thank you very much for your suggestions. In the manuscript, we evaluated the catalytic performance at a relatively low space velocity (2400 L kgcat-1 h-1). Additionally, we also assessed the performance at a higher space velocity (6000 L kgcat-1 h-1); however, due to a significant decline in performance, these results were not adopted in the manuscript. At relatively high space velocities, the differences of catalytic performance became more pronounced, with the borate catalyst AlBOx-1000 exhibiting superior performance.

Figure R1. Propane oxidative dehydrogenation performance of borate catalysts NiBOx-1000 and AlBOx-1000 at higher space velocity: (a) Propane conversion, (b) C3H6 selectivity.

  1. The authors compare the AlBOx-1000 catalyst with AlB2 and state that the former depict superior catalytic performances. AlB2 presents an overall olefin yield of 30.2% at 500°C, however at this temperature (500°C) and with the AlBOx-1000, the propane conversion is barely 10% with an olefin selectivity of ca. 90 %, which gives an olefin yield of less than 10 % - which is much smaller to that of AlB2 at the same temperature – assuming also that both tests were performed under the same reaction conditions (loading, gas florate, O2 content, etc.).

Response: Thank you very much for your suggestions. In the catalytic performance evaluation of AlB2, the feed gas merely contained propane and oxygen. We found that the composition of the feed gas significantly affects the catalytic performance and that radical reactions became more severe as the temperature increases. Consequently, we included an inert gas for dilution in the catalyst performance evaluation. We exclusively compared the maximum olefin yields of the two catalysts and have revised the relevant descriptions accordingly.

  1. The CaBOx-1000 catalyst showed exceedingly low PODH activity, if any, yielding to overoxidation of propane to CO2. Can structural/morphological effects, e.g. the size of the calcium atom (vs Ni and Al), complement the explanations based on electronics. In this regard, textural properties should also be measured for the 3 shaped (pellets) catalysts (N2-physisorption) as used for the catalytic tests.

Response: Thank you very much for your suggestions. We included N2 adsorption-desorption isotherms for the three catalysts and measured their specific surface areas. Our results indicated that there were no significant differences in the specific surface areas of the three catalysts. The differences in catalytic performance were primarily due to the electronegativity of the metals. This phenomenon may be attributed to the low electronegativity of calcium, which increased electron density on the [BO3]3- groups. This, in turn, facilitated the activation of oxygen into more potent oxidative species, leading to excessive oxidation of propane.

Figure R2. N2 adsorption-desorption isotherms of the borate catalysts.

  1. Figure 3d is not discussed

Response: Thank you very much for your suggestions. We have added descriptions to Figure 3d, and the corresponding modifications have been marked in the manuscript.

  1. Figure 4c and 4d obviously refer to the OPDH reaction? to be mentioned in the figure caption

Response: Thank you very much for your suggestions. It is possible that our expression in the figure caption was not clear enough, so we have made the corresponding revisions to the description.

  1. Figure 5 is difficult to comprehend. The authors should draw their tentative reaction mechanism using a central metalloborate element and draw the different reaction steps from propane, together with naming the different steps in link with the description in the main text – this using proper chemical structures

Response: Thank you very much for your suggestions. We have redone the mechanism diagram and labeled the steps in the reaction process.

Figure R3. Possible reaction mechanism on borate catalyst.

  1. Overall, the english phrasing should be improved

Response: We have conducted a thorough review of the entire document and made the necessary modifications.

Reviewer 2 Report

Comments and Suggestions for Authors

This work employed metals (Ni 1.91, Al 1.5 and Ca 1.0) with varying electronegativities to engineer borate catalysts. Among other results, it was found that moderate metal electronegativity favoured propane oxidative dehydrogenation to propylene, whereas excessively low electronegativity led to propane overoxidation to carbon dioxide. Moreover, Aluminum with moderate electronegativity demonstrated optimal performance.  

I found this work interesting. Apart from some typos and grammatical issues, the manuscript reads relatively well, and the results are convincing in terms of characterization, catalytic performance assessment, exploration of the active sites, and reaction mechanisms proposed.

My only concern is that Fig 3d needs a more extended discussion in the manuscript. Including an extended discussion on Fig3d in the main text is crucial for a comprehensive understanding and increases the impact of this work.  Although presented, I think it was never discussed.

The methodology implemented to support the hypothesis looks adequate, in my opinion. Perhaps a deeper discussion of how exactly and what it means to regulate the electron density of boron-oxygen functional groups is needed. A bit more experimental and theoretical support related to the reaction mechanism proposed could also be in order. Perhaps comments on the possible limitations of the proposed reaction mechanism and suggestions on how it can be observed experimentally would be helpful. It is also a good practice to add error bars in the measurements when possible, particularly when discussing catalytic performance.

The oxidative dehydrogenation of propane (ODHP) reaction is important in catalysis and has many applications. The discovery and design of new catalysts are also paramount in this area, and this work focuses on this challenge.

I recommend the publication after addressing my comment.

Comments on the Quality of English Language

The English is fine.

Author Response

Manuscript ID: materials-3028063

Title: Modulating Electron Density of Boron-Oxygen Groups in Borate via Metal Electronegativity for Propane Oxidative Dehydrogenation

Authors: Panpan Li, Yongbin Yao, Shanshan Chai, Zhijian Li, Fan Xue, Xi Wang *

Dear Ms. Diana Alexandra Minea,

Thank you very much for your editorial comments on our manuscript. Firstly, we sincerely appreciate the professional and valuable evaluations and suggestions provided by the editor and reviewers for our work. The insightful and expert comments from the two reviewers have significantly contributed to the enhancement of our work.

We have addressed the comments point by point below. All changes in the revised manuscript were noted with red font. If further information is required, please do not hesitate to contact us.

We hope this manuscript will meet your publication criteria.

Yours sincerely,

Xi Wang

Reviewer 2:

This work employed metals (Ni 1.91, Al 1.5 and Ca 1.0) with varying electronegativities to engineer borate catalysts. Among other results, it was found that moderate metal electronegativity favoured propane oxidative dehydrogenation to propylene, whereas excessively low electronegativity led to propane overoxidation to carbon dioxide. Moreover, Aluminum with moderate electronegativity demonstrated optimal performance.

I found this work interesting. Apart from some typos and grammatical issues, the manuscript reads relatively well, and the results are convincing in terms of characterization, catalytic performance assessment, exploration of the active sites, and reaction mechanisms proposed.

My only concern is that Fig 3d needs a more extended discussion in the manuscript. Including an extended discussion on Fig3d in the main text is crucial for a comprehensive understanding and increases the impact of this work. Although presented, I think it was never discussed.

The methodology implemented to support the hypothesis looks adequate, in my opinion. Perhaps a deeper discussion of how exactly and what it means to regulate the electron density of boron-oxygen functional groups is needed. A bit more experimental and theoretical support related to the reaction mechanism proposed could also be in order. Perhaps comments on the possible limitations of the proposed reaction mechanism and suggestions on how it can be observed experimentally would be helpful. It is also a good practice to add error bars in the measurements when possible, particularly when discussing catalytic performance.

The oxidative dehydrogenation of propane (ODHP) reaction is important in catalysis and has many applications. The discovery and design of new catalysts are also paramount in this area, and this work focuses on this challenge.

I recommend the publication after addressing my comment.

Response: Thank you very much for your suggestions.

Firstly, we have added descriptions to Figure 3d, and the corresponding modifications have been marked in the manuscript.

Secondly, to facilitate a better understanding of the possible reaction mechanism, we have redrawn the mechanism diagram. The current possible reaction mechanism was based on existing characterizations and literatures. To gain a deeper understanding of the potential mechanism, we have designed relevant isotope experiments and integrated theoretical calculations. These studies will be presented in our future work.

Round 2

Reviewer 1 Report

Comments and Suggestions for Authors

Dear authors, thanks for your detailed answers.

Please, could you insert in the SI the N2-physisorption Figure (R2) - and mention these measurments in the manuscript.

Comments on the Quality of English Language

The english was somehow improved - it is not perfect, but understandable

Author Response

Manuscript ID: materials-3028063

Title: Modulating Electron Density of Boron-Oxygen Groups in Borate via Metal Electronegativity for Propane Oxidative Dehydrogenation

Authors: Panpan Li, Yongbin Yao, Shanshan Chai, Zhijian Li, Fan Xue, Xi Wang *

Dear Ms. Diana Alexandra Minea,

Thank you very much for your editorial comments on our manuscript again. We sincerely appreciate the professional and valuable evaluations and suggestions provided by the editor and reviewers for our work.

We have addressed the comments point by point below. All changes in the revised manuscript and supplementary were noted with red font. If further information is required, please do not hesitate to contact us.

We hope this manuscript will meet your publication criteria.

Yours sincerely,

Xi Wang

Reviewer 1:

Please, could you insert in the SI the N2-physisorption Figure (R2) - and mention these measurments in the manuscript.

Response: Thank you very much for your suggestions. We have supplemented the N2-physisorption data in the supporting information and the corresponding description in the manuscript.
